# Impact of Change in Body Composition during Follow-Up on the Survival of GEP-NET

**DOI:** 10.3390/cancers14215189

**Published:** 2022-10-22

**Authors:** Fernando Sebastian-Valles, Nuria Sánchez de la Blanca Carrero, Víctor Rodríguez-Laval, Rebeca Martinez-Hernández, Ana Serrano-Somavilla, Carolina Knott-Torcal, José Luis Muñoz de Nova, Elena Martín-Pérez, Mónica Marazuela, Miguel Antonio Sampedro-Nuñez

**Affiliations:** 1Department of Endocrinology and Nutrition, Hospital Universitario de La Princesa, Instituto de Investigación Sanitaria de La Princesa, 28006 Madrid, Spain; 2Instituto de Investigación Sanitaria del Hospital Universitario de La Princesa, 28006 Madrid, Spain; 3Department of Radiology, Hospital Universitario de La Princesa, 28006 Madrid, Spain; 4Department of General and Digestive Surgery, Hospital Universitario de La Princesa, 28006 Madrid, Spain

**Keywords:** neuroendocrine tumor, body composition, biomarker, prognosis

## Abstract

**Simple Summary:**

Gastroenteropancreatic neuroendocrine tumors (GEP-NETs) are heterogeneous rare diseases causing malnutrition and cachexia in which the study of body composition may have an impact in prognosis. In this context, we aimed to evaluate muscle and fat tissues by computed tomography (CT) at L3 level at diagnosis and at the end of follow-up in a cohort of 98 GET-NET patients and their relationships with clinical and biochemical variables as predictors of survival. Body composition measures and overall mortality correlated with age, ECOG, metastases, LDH, albumin and urea levels. Although there was no relationship between body composition variables at diagnosis and overall and specific mortality, an increase in low-density muscle and a decrease in normal-density muscle during follow-up were independently correlated to overall (*p* < 0.05) and tumor-cause mortality (*p* < 0.05). These results would suggest that nutritional status should therefore be supervised by specialists and an increase in good quality muscle could improve prognosis.

**Abstract:**

Background: Gastroenteropancreatic neuroendocrine tumors (GEP-NETs) are heterogeneous rare diseases causing malnutrition and cachexia in which the study of body composition may have an impact in prognosis. Aim: Evaluation of muscle and fat tissues by computed tomography (CT) at the level of the third lumbar (L3 level) at diagnosis and at the end of follow-up in GET-NET patients and their relationships with clinical and biochemical variables as predictors of survival. Methodology: Ninety-eight GEP-NET patients were included. Clinical and biochemical parameters were evaluated. Total body, subcutaneous, visceral and total fat areas and very low-density, low-density, normal density, high-density, very high-density and total muscle areas were obtained from CT images. Results: Body composition measures and overall mortality correlated with age, ECOG (Eastern Cooperative Oncology Group performance status) metastases, lactate dehydrogenase (LDH), albumin and urea levels. Although there was no relationship between body composition variables at diagnosis and overall and specific mortality, an increase in low-density muscle and a decrease in normal-density muscle during follow-up were independently correlated to overall (*p* <0.05) and tumor-cause mortality (*p* < 0.05). Conclusion: Although body composition measures obtained by CT at diagnosis did not impact survival of GEP-NET patients, a loss of good quality muscle during follow-up was associated with an increased overall and tumor-related mortality. Nutritional status should therefore be supervised by nutrition specialists and an increase in good quality muscle could improve prognosis.

## 1. Introduction

Gastroenteropancreatic neuroendocrine tumors (GEP-NET) constitute a heterogeneous group of neoplasms with a low incidence and a variable behavior [1]. Contrary to most of other types of cancer, the majority of GEP-NET patients usually have mild symptoms and progress over several years. However, some patients rapidly develop a severe disease. Despite different parameters that have been used to date [2,3], there is still no feasible clinical or laboratory index that has been found as particularly useful to predict outcomes [4]. Therefore, identification of early severity markers of mortality or progressive disease, ideally if they were routine tests that can be carried out in all clinical settings, would be especially useful for treatment decisions and follow-up. Several mechanisms have been involved in the relationship between nutrition and the clinical outcomes in oncological disease. Both low and excessive weight have been postulated as risk factors for progression in oncological diseases [4,5,6].

Regarding obesity, current results are conflicting as some patients with obesity are more likely to survive (obesity paradox), hence the BMI (body mass index) is a parameter with limitations in the evaluation of lean tissues [5]. Nowadays, in addition to BMI, we can measure fat and lean tissues to define body composition more precisely in patients with low weight or obesity introducing new concepts related to body composition. In this regard, a loss of muscle mass and muscle strength is defined as sarcopenia and it can be present even in overweight patients. Its development is typical in the elderly, but it is not exclusive [7], and it has been identified as a mortality risk factor independent of BMI in several diseases including oncological disease [8,9,10]. Although there are different pathogenic mechanisms underlying osteosarcopenic obesity and cancer-related sarcopenia [11], these mechanisms can sometimes overlap during the lifetime of a given individual [12].

Different techniques are available to measure body composition (e.g., dual-energy X-ray absorptiometry (DXA), ultrasound [13], bioelectrical impedance analysis [14]). Among all these techniques, computed tomography (CT) images at the level of the third lumbar (L3) have recently been proposed as a widely available method and have been used in the study of body composition in several diseases [15,16,17,18]. Another interesting advantage of this technique is that it allows muscle quality evaluation. In this regard, myosteatosis, defined as poor muscle quality with fat infiltration and lower density, has been also postulated as a mortality risk factor in several diseases including cancer [19].

The prevalence of malnutrition in GEP-NETs patients is not well known to date. However, there are already studies that suggest that patients with GEP-NET have nutritional deficits and that their diet is inadequate [20]. Some studies have evaluated BMI in patients with GEP-NETs, but the association between nutrition and clinical outcomes has not been consistent in these tumors. In addition, the emerging evidence suggests that alterations in body composition can predict the survival of patients with cancer [16,21,22]. In this regard, we provide complete information on body composition from 98 GEP-NET patients including functioning and non-functioning tumors from different locations of the gastrointestinal-pancreatic tract. Our study analyzed body composition measured by L3 CT at diagnosis and at the end of the follow-up and its relation to global and specific mortality. The relationship between body composition and biochemical and clinical parameters was evaluated as a secondary objective. To our knowledge, this is the most extensive GEP-NET cohort evaluating body composition using CT including follow-up that has been published to date.

## 2. Materials and Methods

### 2.1. Recruitment and Variable Collection

This is a single-center, retrospective, observational study including 159 consecutive patients with gastrointestinal and pancreatic NETs. All patients were evaluated in Hospital Universitario de La Princesa between 2004 and 2021. A complete record including history, physical examination and hormone levels was performed in all cases and interpreted by expert endocrinologists in a multidisciplinary team composed of oncologists, surgeons, radiologists and nuclear medicine physicians, classifying all patients according to World Health Organization (WHO) criteria (tumor site and size, angioinvasion, infiltration level, cell proliferation index, immunohistochemical phenotype, and metastases). Patients were managed following current recommendations and guidelines [23,24]. All of them were carefully screened for the presence of other malignancies and genetic disorders. Other endocrine diseases (e.g., Cushing syndrome and acromegaly) were considered as exclusion criteria. We considered overall mortality and also a classification based on the cause: tumor-related mortality, treatment toxicity mortality and others. Patient height, weight and body mass index (BMI) were obtained within one-year pre- and post-CT analysis. All the biochemical variables within the diagnosis year were collected and the mean for each measure was calculated. Those variables with more than 40% of missing values were excluded.

The Internal Ethical Review Committee of the Hospital de La Princesa approved this project. Written informed consent was obtained from all patients before inclusion in accordance with the Declaration of Helsinki.

### 2.2. Analysis of CT Images

An experienced radiologist selected high-quality CT slides of patients at L3 level. Images at diagnosis were used for the analysis. High contrast, unlined images and visible surgeries were exclusion criteria. Two independent experts were trained to perform the processing of CT images with NIH ImageJ version 2.3.0 [25] whose protocol was followed [26,27]. In addition, we coded a macro script to automatically set all the Hounsfield units (HU) required to elucidate tissue areas (Data available in Appendix A and https://github.com/endonutriHUPR/Body-composition-measurement, accessed on 29 September 2022). Then, area subtraction was executed by R version 4.0.3 [28,29]. 

The following body composition measures were obtained: total body area, visceral fat tissue (VAT; HU = −190, −30), subcutaneous fat area (SFA; HU = −190, −30), intermuscular fat area (IMFA; HU = −190, −30), total fat area (TFA; HU = −190, −30), very low-density muscle area (VLDM; HU = −29, −1), low-density muscle area (LDMA; HU = 0, 34), normal-density muscle area (NDMA; HU = 35, 100), high-density muscle area (HDMA; HU = 101, 150), very high-density muscle area (VHDMA; HU = 151, 199) and total muscle area (TMA; HU = −29, 199). After the measurement step, we tested the correlation between both analyses with the Spearman’s rho test. Then, we normalized the data by dividing them by the square of patients’ height in meters and, finally, we obtained the mean of both measures.

In addition, we collected the L3 CT image at the end of the follow-up of each patient defined as death or last available following the same aforementioned statements. We only considered those with at least one year of difference from the diagnosis image. Finally, the difference between final and initial diagnosis images regarding body composition measures was calculated for each patient.

### 2.3. Statistical Analysis

All the statistical analysis was performed using R version 4.0.3 [28]. Normality Shapiro–Wilk test was used to assess normality. Mann–Whitney U or Kruskal–Wallis tests and one-way t-test or ANOVA were used for the evaluation of group differences in those with non-normal and normal distribution, respectively. Spearman’s rho correlation was calculated between continuous variables and body composition measures and a coincident network analysis was done for the estimation of the relationships between qualitative variables (netCoin package [24]) using the median as cut-off point when required. We reported continuous variables as mean ± SD and discrete ones as percentages. Logistic regression was used to include confounding factors with mortality as outcome.

## 3. Results

### 3.1. Cohort Descriptaion

Figure 1 illustrates a flowchart of patient recruitment. From 159 GEP-NET patients, 98 of them (61.64%) were included in the study after applying exclusion criteria at the date of diagnosis. Of these patients, 61 had follow-up CT images. Patients’ mean age was 63.41 ± 15.80 years, and 52.04% were females (Table 1). The most common primary location was small intestine (41.05%) followed by pancreas (37.89%), large intestine (20.00%) and unknown (1.05%) (Table 1). Regarding function: 72.34% were non-functioning, 20.21% functioning carcinoid syndrome, 3.1% gastrinomas and 4.26% insulinomas. The histological classification was G1 37.76%, G2 26.53%, G3 13.27% and 22.45% unknown. Two patients were carriers of MEN1 (Multiple Endocrine Neoplasia Type 1) and one patient of VHL (Von Hippel-Lindau disease). No other apparent genetic abnormalities were found. Mean BMI for the cohort was 25.55 ± 4.6 kg/m^2^. The median follow-up time for patients was 5 years (p25: 2–p75: 8 years) with 33 patients deceased at the end (Table 1).

Regarding body composition, correlation results between the measurements of CT images by both researchers are presented in Appendix A. All the measures displayed a correlation greater than 0.85 except for high-density muscle areas that were removed. Mean values of total fat, subcutaneous fat, visceral fat and intermuscular fat during diagnosis of the 98 patients were 133.8 ± 59.29, 66.92 ± 32.88, 61.35 ± 38.86 and 5.58 ± 3.83 cm^2^/m^2^, respectively (Table 1). Regarding lean muscle, mean values of total muscle, VLD muscle, LD muscle and ND muscle areas were 42.36 ± 7.56, 4.86 ± 2.74; 13.23 ± 5.33 and 23.76 ± 8.00 cm^2^/m^2^, respectively (Table 1).

### 3.2. Variations of Body Composition, Clinical and Biochemical Profile in Relation with Mortality of the Disease

Patients were classified regarding mortality between survivors and non-survivors. The non-survivor group of patients was more likely to be older individuals with a higher prevalence of tumor aggressiveness parameters (Table 2). We found significant associations of survival with age (*p* = 0.010), weight (*p* = 0.020), BMI (*p* = 0.010), tumor grade (*p* < 0.001), ECOG (Eastern Cooperative Oncology Group performance status) (*p* < 0.001), tumor size (*p* = 0.003), presence of metastases and/or residual disease after surgery (*p* < 0.05), and coexistence of non-oncological respiratory diseases (i.e., COPD, *p* < 0.001). When biochemical values were evaluated, we found unfavorable outcomes when patients had higher levels of GOT/AST (*p* = 0.034), LDH (lactate dehydrogenase) (*p* = 0.022) and urea (*p* = 0.001). Interestingly, patients with low albumin (*p* = 0.016) also displayed a worse survival rate. Other biochemical variables did not reach significant results (Table 2).

Regarding associations between mortality and body composition at diagnosis, the non-survivor group had significantly different baseline profiles than the survivor group (Table 2). The non-survivor group had lower levels of total muscle area at diagnosis (mean survival group = 43.83 cm^2^/m^2^, mean mortality group = 39.18 cm^2^/m^2^, *p* = 0.005), a decrease in muscle density (mean survival group = 25.15 cm^2^/m^2^, mean mortality group = 20.76 cm^2^/m^2^, *p* = 0.007) and lower levels in subcutaneous fat (mean survival group = 72.25 cm^2^/m^2^, mean mortality group = 55.25 cm^2^/m^2^, *p* = 0.008). In addition, we found an increasing tendency in TFA (mean survival group = 142.03 cm^2^/m^2^, mean mortality group = 117.47 cm^2^/m^2^, *p* = 0.052) in relation to survival (Table 2). No differences were observed for visceral fat, intermuscular fat, very low and low-density muscle composition parameters. 

### 3.3. Correlations between Body Composition and Clinico-Biochemical Variable in Relation to Survival

When exploring in depth the relationships among body composition, clinical and biochemical variables and survival, Spearman’s rho analysis (Figure 2A) showed strong positive correlations between low-density muscle and the different fat compartments that were measured (0.485 < r < 0.754, *p* < 0.001). Furthermore, there is an inverse correlation between NDMA and low-density muscle, particularly intense with VLDMA (r = −0.750, *p* < 0.001), and fat compartments, especially with IMFA (r = −0.734, p = 0). Regarding TMA, moderate positive correlations with NMDA (r = 0.456, *p* < 0.001), LDMA (r = 0.487, *p* < 0.001), VAT (r = 0.421, *p* < 0.001) and body area (r = 0.320, *p* = 0.006) were found.

Interestingly, we found negative correlations between NDMA with LDH (r = −0.442, *p* < 0.001), neutrophils percentage (r = −0.240, *p* = 0.048) and age (r = −0.547, *p* < 0.001). In contrast, NMDA had a weak positive correlation with lymphocytes percentage (r = 0.291, *p* = 0.016) (Figure 2A). In addition, weak positive correlations were present between LDMA and fibrinogen (r = 0.361, *p* = 0.004), glucose (r = 0.250, *p* = 0.040) and triglycerides (r = 0.338, *p* = 0.007).

Finally, a network coincidence analysis was performed between qualitative variables to elucidate shared relationships between survival-mortality and body composition measures with clinical parameters. Significant correlations are represented in Figure 2B. There were relationships among survival and mortality conditions with SFA, TFA and total area and ECOG scale (Haberman residual > 1.78 and <4.63, *p* < 0.05), metastases (Haberman residual >2.21 and <3.28, *p* < 0.05), nodules (Haberman residual >1.69 and <2.26, *p* < 0.05), and non-functioning tumors (Haberman residual > 1.79 and <2.57, *p* < 0.05). Interestingly, the use of somatostatin analogues or diabetes diagnosis did not reach significant results in body composition and mortality (Table 2 and Appendix A).

After all the results, we underlined age, metastasis, ECOG, LDH, albumin and urea as confounding factors in the relationship between body composition and future mortality. When confounding factors were added in the multivariable analysis, the effect of baseline body composition at diagnosis on survival was not significant (neither a risk nor a protective factor) (Figure 3).

### 3.4. Body Composition Changes during Follow-Up Have an Impact on Mortality

Therefore, we evaluated the changes in body composition at the end of the follow-up (5 years median, p25: 2–p75: 8 years) and their impact in overall mortality through logistic regression analysis using the confounding factors previously identified. Results are displayed in Table 3 and Figure 4A. There was a general decrease in total muscle mass during follow-up (from 41.67 cm^2^/m^2^ to 39.07 cm^2^/m^2^) with no differences when compared between survival and mortality (*p* = 0.334). Interestingly, patients with adverse outcomes gained low-density muscle (from 12.07 cm^2^/m^2^ to 13.41 cm^2^/m^2^, *p* = <0.05) and lost normal-density muscle (from 21.19 cm^2^/m^2^ to 16.01 cm^2^/m^2^, *p* < 0.05) at the end of the follow-up (Figure 4). Despite a severe loss of fat tissue in patient’s death, adjusted parameters showed a non-significant relationship (from 109.30 cm^2^/m^2^ to 90.95 cm^2^/m^2^, *p* = 0.680).

We then performed the analyses specifically in tumor-cause mortality; their results maintained the tendency that we previously observed in all-cause mortality (Table 3). There was an increment in low-density muscle area (*p* < 0.05) and a decrease in normal-density muscle area (*p* < 0.05) compared to survival. Changes are represented in Figure 4B,C.

These results seemed to advise against the importance of muscle quality in GEP-NET patients regarding its relationship with overall and tumor-cause mortality during the follow-up.

## 4. Discussion

Emerging evidence suggests that alterations in body composition can predict the survival of patients with cancer [16,18,21,22]. In this regard, we analyzed body composition measured by L3 CT in 98 patients. 

When evaluating patient outcomes in univariate analysis, we found that the amount of muscle and fat compartments and also its quality and fat distribution at diagnosis were related to patient outcomes. Specifically, we found that less total fat, visceral fat, subcutaneous fat, less total muscle, and decreased muscle quality were associated with increased mortality risk. When relationships were assessed, an increase in muscle area of normal-density was negatively correlated with low-density muscle area and, in consequence, with myosteatosis and fat tissue. However, after correction with confounding factors, we observed that the study of body composition using computed tomography technology at diagnosis had no independent correlations with survival of patients with GEP-NETs. Our results are consistent with previous works with the same disease that studied the relationship between body composition and unfavorable outcomes. Sarcopenia was not associated with long-term prognosis, but was an independent risk factor in a small subgroup of patients with gastric mixed adenoneuroendocrine carcinoma (gMANEC) and an independent risk factor after surgery [30]. In another study that evaluated the tumor-ghrelin system and body composition at diagnosis in 63 patients, weight loss at diagnosis and sarcopenia were not statistically significant in relation to higher mortality [31]. Nicoletta Ranallo et al. analyzed only metastatic neuroendocrine tumors before and after being treated with everolimus and showed no change in muscle mass and progression. They did not find a correlation with fat and treatment response [32]. In accordance with these scientific works, Chan et al. also concluded in a cohort of neuroendocrine neoplasms treated with peptide receptor radionuclides that sarcopenia and myosteatosis at the start of treatment (muscle quantity and quality) were not correlated with subsequent survival [33].

However, during the follow-up of the body composition in our cohort of patients, the results in survival were very interesting and showed that both the loss of normal-density muscle and the gain of low-density muscle were independently related to all-cause mortality. Additionally, when studying tumor-related mortality, we observed that the relationship with the loss of muscle of normal density and increase of muscle of low density was maintained. These changes were adjusted by age, presence of metastases, ECOG, LDH, albumin, and urea. Studies are currently being carried out to assess muscle maintenance as a variable related to survival. In a study of 18 patients with malignant mesothelioma, muscle loss during follow-up measured by DXA was associated with decreased survival [34]. 

The muscle density loss in patients with NETs could be due to several factors, including paraneoplastic syndromes, hormonal hypersecretion, including the physical presence of cancer, mechanical symptoms related to the disease or treatment, and malabsorption [35]. All these factors can be included in the cancer-related cachexia syndrome. This term describes a phenomenon that results from a combination of reduced food intake and metabolic changes including increased energy expenditure, excessive catabolism, inflammation and age [36]. Inflammation takes on particular weight in the onset of sarcopenia and myosteatosis. Tumors secrete molecules that cause further catabolism in the tissues [37]. Classical proinflammatory factors that produce catabolic actions have been extensively studied as mediators of cachexia [38]. These signaling molecules are synthesized by tumors or immune cells and their activities are sufficient to promote catabolism in target organs such as skeletal muscle [33]. Elevated fibrinogen levels have been described as a proinflammatory factor that acts as an independent factor for all-cause mortality in other cancer patients [39]. Multiple proinflammatory effector pathways also modulate homeostatic controls in the CNS, eliciting neural and neuroendocrine responses. These homeostatic controls promote adrenal steroid release and alter disease behavior leading to anorexia and fatigue. Through these humoral, neural, and behavioral outcomes, lipolysis and proteolysis are directly activated in muscle, heart, and adipose tissue [37].

In our cohort, a higher volume of normal muscle was an independent factor of survival in the follow-up, and it was positively correlated with albumin and lymphocytes and negatively correlated with age at diagnosis. On the contrary, increase of low-density muscle tissue (myosteatosis) had a relationship with mortality in the follow-up, and were positively related to higher levels of fibrinogen, LDH, glucose, and triglycerides at diagnosis, parameters that were previously correlated with poor prognosis [40,41]. In addition, there were other associations with body composition and bad prognosis characteristics such as metastasis, higher ECOG and advanced age, also associated with mortality.

The low-density muscle gain is due to a loss of normal muscle. Therefore, weight gain does not have an impact on mortality if muscle quality is not improved. Nutritional management during the therapeutic itinerary is vital in patients with GEP-NET. For this reason, hopefully, a modifiable factor such as a change in body composition during follow-up may influence the prognosis of patients. Still, more extensive studies are needed to clarify its real potential in clinical practice and which strategies measures (diet, immunonutrients and/or exercise) might enhance the quality of muscle during follow-up. In this regard, the collaboration between multidisciplinary teams that include oncologists, endocrinologists and nutritionists are necessary to identify patients at high risk and to implement physical exercise and specific nutrition prevention programs. In this context, the research team agrees with the recommendations previously published by Kikut et al. [20] for the nutritional management of patients with neuroendocrine tumors where it is proposed that standardizing nutritional care of patients with NETs should be a priority direction in the management of these patients. Due to the heterogeneity of this oncological pathology, the nutritional management of each patient will probably have to be individualized. Still, the research team agrees with the initial recommendation proposed by the ESPEN 2020 guidelines for oncological patients [42] where it is established that all patients with oncological pathology must be evaluated, recording parameters such as weight loss or, for instance, body composition as proposed in our study. The timing of this evaluation should be before starting treatment at diagnosis, and these measurements should be repeated depending on the stability of the patient’s clinical situation. In addition, the complementary use of other measures, such as the phase angle measured by electric bioimpedance of the anterior rectum of the quadriceps [43], will allow us in the future to stratify patients and probably select different oncologic treatment strategies based on their frailty condition.

Finally, one of the main limitations of this study is the observational methodology of the work. Another aspect to consider is that the cohort is part of a single center, which can bias the reproducibility of the study. On the other hand, the cohort is one of the largest published, and the long follow-up period from 2004 to 2021 offers advantages in providing information on patient’s natural history. Still, the appearance of new treatments and the specific effect of any treatment in body composition have not been discussed in this paper and should be evaluated in future studies. Other aspects to consider in future studies are muscle strength (i.e., hand grip strength) and physical activity information of patients at diagnosis and during the follow-up.

For all these reasons, multicenter clinical trials should be conducted to better understand the influence of body composition in patients with GEP-NET. Early nutritional treatment with particular emphasis on maintaining and increasing muscle mass should be a priority in the follow-up of patients with GEP-NET. Therefore, it seems necessary to implement therapies that allow our patients to maintain or increase muscle quality as a primary objective.

## 5. Conclusions

Body composition analysis is feasible using CT data acquired in routine clinical practice in patients with NETs. Loss in normal-density muscle and increase in low-density muscle during follow-up are independently associated with an increase in mortality. Nutritional management during the follow-up should be mandatory for patients with GEP-NET.

## Figures and Tables

**Figure 1 cancers-14-05189-f001:**
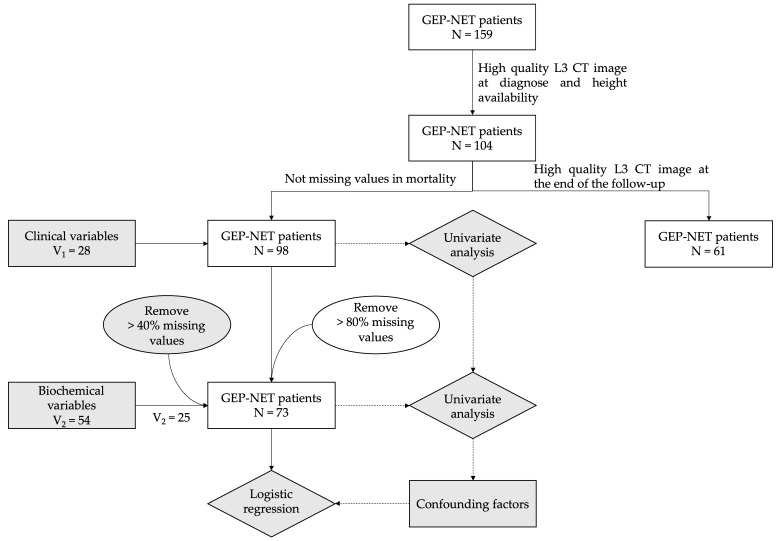
Recruitment strategy flow chart. Patients and variables selection from hospital database. GEP-NET, gastroenteropancreatic tumors; CT, computed tomography.

**Figure 2 cancers-14-05189-f002:**
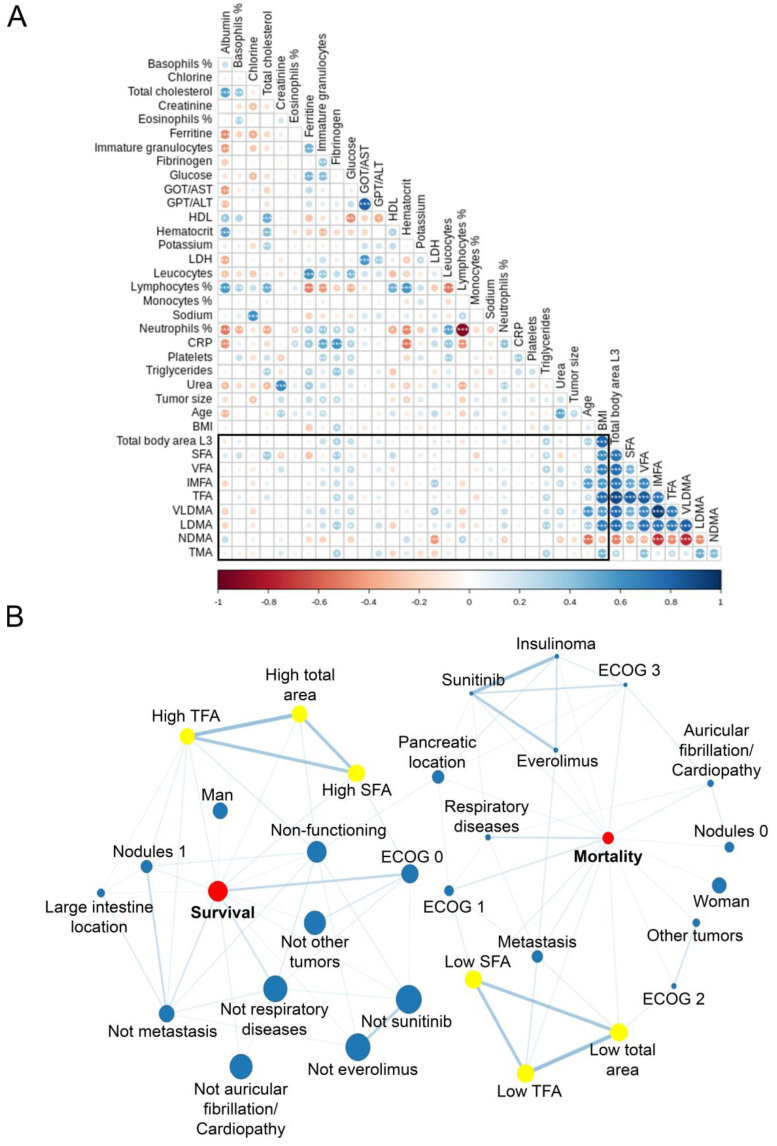
Relationships between body composition measures and clinical and biochemical variables. (**A**) Spearman’s rho correlation matrix between body composition at L3 CT images and continuous variables. Blue intensity corresponds to positive correlations whereas red one indicates those that were negative. * *p* < 0.05; ** *p* <0.01; *** *p* <0.001. (**B**) The coincidence network with Haberman residues. Node sizes correlate with category percentage and link width and color intensity correlate with Haberman coefficients. Only significant (*p* < 0.05) relationships were represented. CT, computed tomography; GOT/AST, glutamyl oxaloacetic transaminase/aspartate aminotransferase; GPT/ALT, glutamyl pyruvic transaminase/alanine aminotransferase; HDL, high-density lipoprotein; LDH, lactate dehydrogenase; CRP, C reactive protein; SFA, subcutaneous fat area; VFA, visceral fat area; IMFA, intermuscular fat area; TFA, total fat area; VLDMA, very low-density muscle area; LD, low-density muscle area; NDMA, normal-density muscle area; TMA, total muscle area.

**Figure 3 cancers-14-05189-f003:**
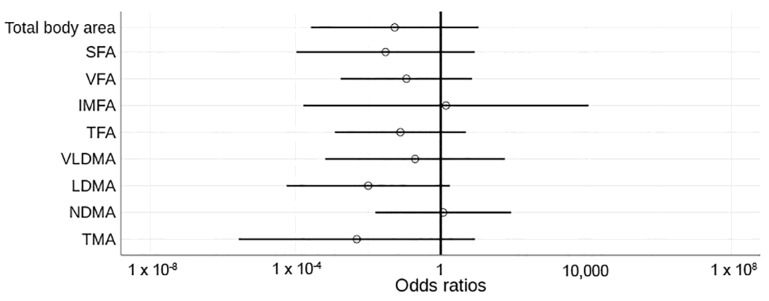
Odd ratios of body composition measures and their relationships with overall mortality. Unfilled figures indicate non-significant values. Vertical black line highlights odd ratio = 1. Confounding factors in the model: age, metastases, ECOG (Eastern Cooperative Oncology Group performance status), lactate dehydrogenase (LDH), albumin and urea. SFA, subcutaneous fat area; VFA, visceral fat area; IMFA, intermuscular fat area; TFA, total fat area; VLDMA, very low-density muscle area; LD, low-density muscle area; NDMA, normal-density muscle area; TMA, total muscle area.

**Figure 4 cancers-14-05189-f004:**
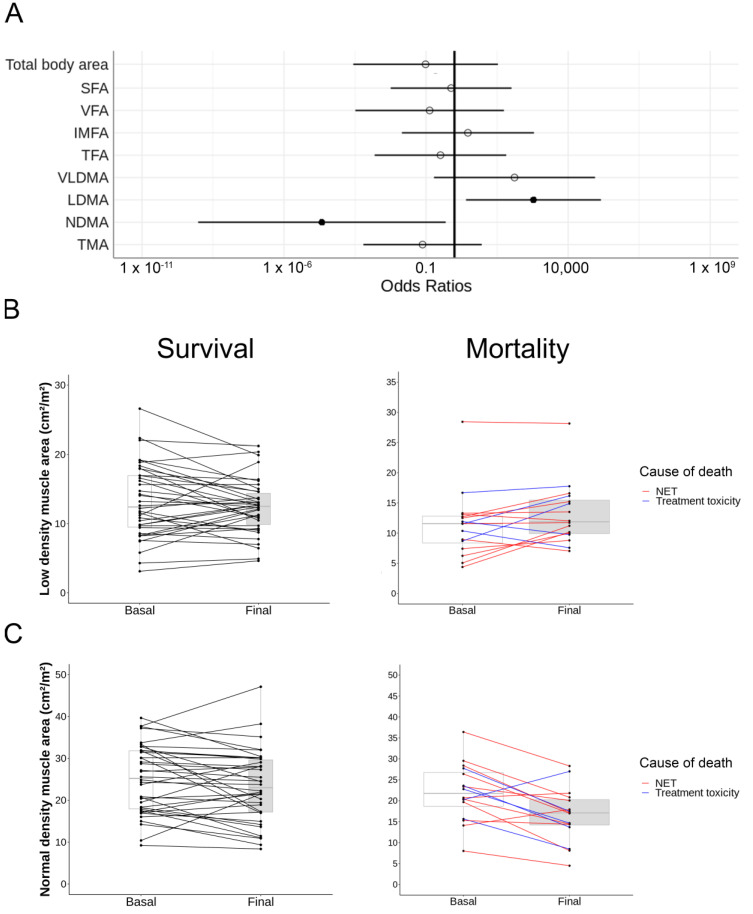
Changes in body composition and their relationships with mortality. (**A**) Odds ratio of body composition measures and relationship with overall mortality. Filled circles represent significant protective (left) and risk factors (right) of overall mortality in this cohort, whereas unfilled figures indicate non-significant values. Vertical black line highlights odd ratio = 1. Confounding factors in the model: age, metastases, ECOG (Eastern Cooperative Oncology Group performance status), lactate dehydrogenase (LDH), albumin and urea. SFA, subcutaneous fat area; VFA, visceral fat area; IMFA, intermuscular fat area; TFA, total fat area; VLDMA, very low-density muscle area; LD, low-density muscle area; NDMA, normal-density muscle area; TMA, total muscle area. (**B**) Change in low-density muscle and (**C**) normal-density muscle areas in survival (left) and mortality (right). Black lines correspond to survival patients, red lines to tumor-related mortality and blue ones to treatment-toxicity-related mortality.

**Table 1 cancers-14-05189-t001:** Cohort description.

Variables	Overall N = 98
Age (years), mean (SD)	63.34 (15.74)
Sex (M/F), N (%)	50 (51.02%)/48 (48.98%)
Tumor location, N (%)	Small intestine: 34 (40.00%) Pancreas: 37 (38.95%) Large intestine: 19 (20.00%) Undefined: 1 (1.05%)
Exitus, N (%)	33 (33.67%)
BMI (kg/m^2^), mean (SD)	25.55 (4.60)
Total area (cm^2^/m^2^), mean (SD)	262.09 (64.20)
Subcutaneous fat area (cm^2^/m^2^), mean (SD)	66.92 (32.88)
Visceral fat area (cm^2^/m^2^), mean (SD)	61.35 (38.86)
Intermuscular fat area (cm^2^/m^2^), mean (SD)	5.58 (3.83)
Total fat area (cm^2^/m^2^), mean (SD)	133.84 (59.29)
VLD muscle area (cm^2^/m^2^), mean (SD)	4.86 (2.74)
LD muscle area (cm^2^/m^2^), mean (SD)	13.23 (5.33)
ND muscle area (cm^2^/m^2^), mean (SD)	23.76 (8.00)
Total muscle area (cm^2^/m^2^), mean (SD)	42.36 (7.56)

Continuous and categorical data are presented as mean with standard deviation (SD) and percentage, respectively. VLD, very low-density; LD, low-density; ND, normal density.

**Table 2 cancers-14-05189-t002:** Biochemical, clinical and body composition variables and their correlations with survival and overall mortality.

Variable	Overall (N = 98)	Survival (N = 65)	Mortality (N = 33)	*p* Value
**General clinical variables at diagnose**				
Age *	63.41 (15.80)	60.58 (15.63)	68.97 (14.82)	0.01
Female	51 (52.04%)	37 (56.92%)	14 (42.42%)	0.25
Never smoked	49 (57.65%)	35 (64.91%)	14 (51.85%)	0.55
Currently smoking	17 (20.00%)	11 (19.30%)	6 (22.22%)	
Non currently smoking	19 (22.35%)	11 (19.30%)	7 (25.93%)	
Weight +	70.53 (16.71)	74.49 (19.19)	64.69 (9.97)	0.02
BMI +	25.85 (4.65)	26.64 (5.16)	24.31 (2.96)	0.01
**Tumor clinical variables**				
Grade (G)				<0.01
0	2 (2.56%)	1 (1.49%)	1 (4.76%)	
1	37 (47.44%)	33 (49.25%)	4 (19.05%)	
2	26 (33.33%)	20 (29.85%)	6 (28.57%)	
3	11 (14.10%)	3 (4.48%)	8 (38.09%)	
4	2 (2.56%)	0	2 (2.52%)	
ECOG 0	57 (60.00%)	48 (77.42%)	9 (27.27%)	<0.01
ECOG 1	27 (28.42%)	11 (17.74%)	16 (48.48%)	
ECOG 2	8 (8.42%)	3 (4.84%)	5 (15.15%)	
ECOG 3	3 (3.16%)	0	3 (9.09%)	
Carcinoid Syndrome	19 (20.21%)	11 (17.19%)	8 (26.67%)	0.19
Gastrinoma	3 (3.19%)	2 (3.12%)	1 (3.33%)	
Insulinoma	4 (4.26%)	1 (1.56%)	3 (10.00%)	
Tumor size (mm) *	40.25 (43.05)	28. 94 (30.72)	62.88 (54.48)	0.01
Functioning	9 (11.39%)	5 (9.80%)	4 (14.29%)	0.82
Residual tumor:				0.04
0	45 (48.91%)	37 (57.81%)	8 (28.57%)	
1	37 (40.22%)	21 (32.82%)	16 (57.14%)	
NA	10 (10.87%)	6 (9.38%)	4 (14.29%)	
Location of primary tumor:				0.23
Small intestine	39 (41.05%)	27 (41.54%)	12 (40.00%)	
Pancreatic	36 (37.89%)	21 (32.31%)	15 (50.00%)	
Large intestine	19 (20.00%)	16 (24.62%)	3 (10.00%)	
Undefined	1 (1.05%)	1 (1.54%)	0	
Nodules				0.12
0	23 (27.38%)	13 (20%)	10 (30.303%)	
1	31 (36.9%)	26 (40%)	5 (15.152%)	
2	21 (25%)	15 (23.077%)	6 (18.182%)	
3	8 (9.52%)	5 (7.692%)	3 (9.091%)	
Nx	1 (1.19%)	0 (0%)	1 (3.03%)	
Unknown	10 (11.9%)	5 (7.692%)	5 (15.152%)	0.01
No metastasis	45 (60.81%)	38 (71.70%)	7 (33.33%)	
Metastasis	29 (39.19%)	15 (28.30%)	14 (66/67%)	
Incidental NET	50 (52.63%)	33 (51.56%)	17 (54.84%)	0.94
**Body composition in L3 CT images**				
Body area +	262.05 (64.24)	269.19 (66.28)	247.99 (58.43)	0.11
SFA +	66.59 (33.12)	72.25 (35.25)	55.45 (25.44)	0.01
VFA *	61.55 (38.72)	64.34 (36.95)	56.06 (42.03)	0.15
TFA +	133.76 (59.36)	142.03 (58.94)	117.47 (57.62)	0.05
IMFA *	5.61 (3.82)	5.44 (4.12)	5.95 (3.17)	0.24
VLDMA *	4.88 (2.73)	4.72 (2.65)	5.20 (2.89)	0.49
LDMA +	13.27 (5.29)	13.55 (5.26)	12.71 (5.39)	0.46
NDMA +	23.67 (8.02)	25.15 (8.17)	20.76 (6.96)	0.01
TMA *	42.33 (7.59)	43.93 (7.37)	39.18 (7.11)	<0.01
**Drugs–Treatment**				
Metformin	17 (19.10%)	13 (21.67%)	4 (13.79%)	0.25
Somatostatin Analogues	32 (32.65%)	20 (30.77%)	12 (36.36%)	0.74
Everolimus	6 (6.12%)	2 (3.08%)	4 (12.12%)	0.19
Sunitinib	2 (2.04%)	0	2 (6.06%)	0.21
Radionuclides	8 (8.16%)	5 (7.69%)	3 (9.09%)	1.000
**Other diseases at diagnosis**				
Arterial hypertension	47 (51.65%)	30 (50.00%)	17 (54.84%)	0.83
Diabetes Mellitus	34 (35.05%)	23 (35.38%)	11 (34.38%)	0.77
Other tumors	18 (19.78%)	8 (13.33%)	10 (32.26%)	0.06
Atrial fibrillation/Cardiopathy	13 (14.29%)	5 (8.33%)	8 (25.81%)	0.05
Cardiovascular disease	13 (14.29%)	6 (10.00%)	7 (22.58%)	0.19
Respiratory disease	10 (11.11%)	1 (1.69%)	9 (29.03%)	<0.01
Reuma	11 (12.22%)	6 (10.17%)	5 (16.13%)	0.55
Infectious disease	3 (3.33%)	2 (3.39%)	1 (3.23%)	0.77
Autoimmune disease	9 (10.00%)	5 (8.47%)	4 (12.90%)	0.62
**Biochemical variables at diagnosis**				
Albumin *	3.98 (0.61)	4.09 (0.58)	3.71 (0.61)	0.02
Fibrinogen +	537.37 (155.60)	551.67 (154.64)	504.26 (156.86)	0.28
Glucose *	117.37 (30.44)	116.98 (31.43)	118.26 (28.74)	0.86
Urea *	37.85 (13.77)	34.49 (11.00)	45.21 (16.44)	0.01
Creatinine *	0.90 (0.39)	0.84 (0.25)	1.04 (0.59)	0.06
Sodium +	139.85 (2.24)	140.03 (1.86)	139.47 (2.92)	0.43
Potassium +	4.28 (0.32)	4.27 (0.27)	4.29 (0.41)	0.91
GOT/AST *	31.74 (36.72)	22.94 (6.78)	52.85 (63.05)	0.04
GPT/ALT *	32.86 (31.05)	28.64 (19.26)	42.97 (48.26)	0.52
LDH *	206.04 (79.70)	182.53 (39.31)	255.55 (115.35)	0.02
Cholesterol +	167.30 (46.66)	172.08 (46.89)	156.33 (45.38)	0.21
HDL +	44.22 (14.91)	45.04 (15.71)	42.70 (13.68)	0.62
Triglycerides *	128.01 (71.10)	135.78 (82.20)	110.90 (31.96)	0.52
Ferritine *	256.66 (509.75)	155.67 (139.52)	416.13 (795.94)	0.14
CRP *	3.60 (3.91)	3.50 (3.93)	3.83 (3.97)	0.56
Hematocrit +	4.32 (0.67)	4.41 (0.63)	4.14 (0.73)	0.15
Leucocytes *	8.87 (3.11)	8.79 (2.79)	9.06 (3.82)	0.46
Lymphocytes (%) +	20.49 (9.82)	21.30 (9.38)	18.62 (10.76)	0.33
Monocytes (%) +	8.11 (2.20)	8.02 (1.98)	8.33 (2.68)	0.64
Neutrophils (%) +	67.28 (10.52)	66.96 (9.52)	68.01 (12.74)	0.74
Immature granulocytes (%) *	0.42 (0.34)	0.43 (0.38)	0.41 (0.22)	0.66
Eosinophils (%) *	2.16 (1.43)	2.10 (1.39)	2.31 (1.55)	0.54
Basophils (%) *	0.42 (0.23)	0.43 (0.23)	0.40 (0.21)	0.50
Platelets *	257.30 (96.34)	264.74 (99.90)	240.29 (87.54)	0.27

NET, neuroendocrine tumors; BMI, body mass index; ECOG, Eastern Cooperative Oncology Group performance status; SFA, subcutaneous fat area; VFA, visceral fat area; IMFA, intermuscular fat area; TFA, total fat area; VLDMA, very low-density muscle area; LD, low-density muscle area; NDMA, normal-density muscle area; TMA, total muscle area; NA, not available, GOT/AST, glutamyl oxaloacetic transaminase/aspartate aminotransferase; GPT/ALT, glutamyl pyruvic transaminase/alanine aminotransferase; HDL; high-density lipoprotein; LDH, lactate dehydrogenase; CRP, C reactive protein; RDW, red blood cell distribution width. * Mann–Whitney U test; + *t*-test analysis; Chi-squared analysis does not have any symbol.

**Table 3 cancers-14-05189-t003:** Logistic regression output correcting follow-up body composition measures and their distribution between survival and overall mortality and survival and tumor-cause-related mortality.

Body Composition Measures	Status	Overall Mortality	*p*-Value *	Tumor-Cause Mortality	*p*-Value *
Total area (cm^2^/m^2^), mean (SD)	Survival	−0.88 (35.85)	0.36	0.89 (37.01)	0.44
	Mortality	−16.16 (50.37)		−1.90 (19.27)	
Subcutaneous fat area (cm^2^/m^2^), mean (SD)	Survival	−0.47 (17.05)	0.85	0.44 (17.75)	0.32
	Mortality	−11.09 (25.49)		−10.46 (14.85)	
Visceral fat area (cm^2^/m^2^), mean (SD)	Survival	−0.03 (21.23)	0.56	0.74 (21.47)	0.44
	Mortality	−10.97 (32.19)		−3.75 (20.96)	
Intermuscular fat area (cm^2^/m^2^), mean (SD)	Survival	1.06 (2.36)	0.74	1.13 (2.37)	0.76
	Mortality	0.76 (2.91)		1.04 (2.43)	
Total fat area (cm^2^/m^2^), mean (SD)	Survival	0.56 (35.39)	0.68	2.32 (36.5)	0.37
	Mortality	−21.30 (54.02)		−13.17 (30.40)	
VLD muscle area (cm^2^/m^2^), mean (SD)	Survival	0.16 (1.51)	0.24	0.18 (1.50)	0.10
	Mortality	1.13 (2.41)		1.63 (2.33)	
LD muscle area (cm^2^/m^2^), mean (SD)	Survival	−0.92 (3.60)	0.02	−0.83 (3.59)	0.03
	Mortality	1.22 (3.54)		1.89 (2.75)	
ND muscle area (cm^2^/m^2^), mean (SD)	Survival	−1.61 (5.85)	0.01	−1.72 (5.81)	0.05
	Mortality	−4.96 (5.07)		−5.25 (5.11)	
Total muscle area (cm^2^/m^2^), mean (SD)	Survival	−2.43 (3.82)	0.33	−2.43 (3.77)	0.72
	Mortality	−2.60 (5.72)		−3.15 (6.48)	

Data are presented as mean with standard deviation (SD). VLD, very low-density; LD, low-density; ND, normal density. * Adjusted by: age, metastasis, ECOG (Eastern Cooperative Oncology Group performance status), lactate dehydrogenase (LDH), albumin and urea.

## Data Availability

Data available upon reasonable request.

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
