# Peer review of "Impact of Change in Body Composition during Follow-Up on the Survival of GEP-NET"

_cancers, 2022, doi:10.3390/cancers14215189_

Round 1

Reviewer 1 Report

The paper is focused on a very interesing topic as only few data are avalaible in literature. However, there are some limitations which don't allow o draw firm conclusions, the main one being he lack of data regarding the specific effect of any treatment in this cohort of patients. Some considerations:

- I'd suggest to improve the introduction by adding some data regarding the nutritional status of patients with GEP-NENs including macro- and micronutrient deficiencies (i.e. Nutrients. 2020 Jun 30;12(7):1961)

- Materials and Methods: The authors state that the data were recorded and interpreted by expert endocrinologists, but I guess the patients were all managed by a multidisciplinary team, thus I'd suggest to edit it otherwise it migh be misguiding

- Figure 1. GEP NET patients = 104, but in he text (page 2) the authors state 105 consecutive patients. Please clarify.

- In the discussion and conclusion the authors highlight the fundamental role of a multidisciplinary management together with a nutritional management. I'd suggest the authors to try to provide a practical message on how the patients should be managed from a nutritional point of view (specific exams to be requested, timing of nutritional assessment etc). 

- English language should be improved

Author Response

Dear reviewer,

Thank you very much for all the suggestions you have made. In this letter, we respond to each of them. Changes made to the manuscript are marked in red.

REVIEWER 1:

The paper is focused on a very interesting topic as only few data are available in literature. However, there are some limitations which don't allow to draw firm conclusions, the main one being the lack of data regarding the specific effect of any treatment in this cohort of patients. Some considerations:

Indeed, that could be a limitation because of the treatment diversity, their possible independent relationships with body composition and the N of the study, having a moderate statistical power to evaluate it. In addition to that, Nicoletta Ranallo et al published a manuscript evaluating the effect of everolimus in body composition and progression and  did not find changes in muscle related to this treatment.

Within this context, classical statistics is not trustworthy due to the limited number of patients: (everolimus N=6, sunitinib N = 2,  radionuclides N=9). In addition and in spite of drugs such as everolimus and sunitinib appearing in the coincidence network with mortality and metastasis (Figure 2), we could not consider them as confounding factors since they do not have relationships to any of the body composition features we measured (Figure 2). We could only provide you with confidence a classical analysis of metformin and somatostatin analogues (below). We noticed that somatostatin analogues were correlated to fat tissue features,  but not to mortality/survival (p = 0.74). The same occurred with metformin: it had correlations with fat tissue and low density muscle area, but not to mortality/survival (p = 0.25). Due to this fact, we do not consider them as confounding factors in the relationships between body composition and an unfavorable outcome, which was the objective of the study.

Nevertheless, in line to your assessment, we would suggest a wider study focusing on the effect of these drugs to address this limitation. We better explained this issue in the limitations paragraph of the discussion to leave the door open to future studies in the area in page 14, line 416. We have added this table to supplementary material S2 and added results in page 10, line 269.

Measure (cm²/m²)

Overall

Not somatostatin analogues

Somatostatin analogues

p value

Test

Total body area

262.085 (64.196)

270.093 (68.595)

245.569 (51.099)

0.0508

T-test

Subcutaneous fat area

66.918 (32.882)

69.68 (35.727)

61.223 (25.631)

0.3899

Mann- Whitney wilcoxon

Visceral fat area

61.346 (38.857)

66.928 (39.165)

49.834 (36.127)

0.0168

Mann- Whitney wilcoxon

Intermuscular fat area

5.579 (3.825)

6.1 (4.181)

4.505 (2.714)

0.0272

Mann- Whitney wilcoxon

Total fat area

133.844 (59.289)

142.709 (60.824)

115.562 (52.245)

0.0254

T-test

Very low density muscle area

4.864 (2.74)

5.207 (2.903)

4.156 (2.249)

0.0567

Mann- Whitney wilcoxon

Low density muscle area

13.225 (5.331)

13.674 (5.335)

12.297 (5.285)

0.2324

T-test

Normal density muscle area

23.759 (8.002)

23.206 (7.526)

24.899 (8.922)

0.359

T-test

Total muscle area

42.36 (7.561)

42.626 (7.326)

41.812 (8.117)

0.6327

T-test

Measure (cm²/m²)

Overall

Not metformin

Metformin

p value

Test

Total body area

262.085 (64.196)

252.89 (60.527)

298.124 (60.589)

0.0107

T-test

Subcutaneous fat area

66.918 (32.882)

67.224 (33.545)

72.703 (36.294)

0.5762

T-test

Visceral fat area

61.346 (38.857)

53.603 (31.685)

81.591 (45.552)

0.0319

Mann- Whitney wilcoxon

Intermuscular fat area

5.579 (3.825)

5.432 (4.052)

5.796 (3.042)

0.4404

Mann- Whitney wilcoxon

Total fat area

133.844 (59.289)

126.26 (56.53)

160.089 (59.544)

0.0444

T-test

Very low density muscle area

4.864 (2.74)

4.575 (2.518)

5.697 (3.217)

0.2085

Mann- Whitney wilcoxon

Low density muscle area

13.225 (5.331)

12.478 (4.772)

15.922 (5.675)

0.0305

T-test

Normal density muscle area

23.759 (8.002)

24.386 (8.135)

20.896 (7.603)

0.1058

T-test

Total muscle area

42.36 (7.561)

41.979 (7.467)

42.924 (8.761)

0.5191

Mann- Whitney wilcoxon

- I'd suggest to improve the introduction by adding some data regarding the nutritional status of patients with GEP-NENs including macro- and micronutrient deficiencies (i.e. Nutrients. 2020 Jun 30;12(7):1961).

Thank you very much for the suggestion. We have added and quoted the information provided in the paper (page 2, line number 88).

- Materials and Methods: The authors state that the data were recorded and interpreted by expert endocrinologists, but I guess the patients were all managed by a multidisciplinary team, thus I'd suggest to edit it otherwise it migh be misguiding.

The treatment of patients with GEP-NET is carried out by a multidisciplinary team composed of oncologists, surgeons, radiologists, nuclear physicians and endocrinologists. We are sorry for the mistake and we have specified this aspect in the text (page 3, line number 108).

- Figure 1. GEP NET patients = 104, but in he text (page 2) the authors state 105 consecutive patients. Please clarify.

We apologize for the misunderstanding. N changes due to inclusion/exclusion criterias were not clearly detailed in the text. We have written a better explanation now (page 4, line number 165).

- In the discussion and conclusion the authors highlight the fundamental role of a multidisciplinary management together with a nutritional management. I'd suggest the authors to try to provide a practical message on how the patients should be managed from a nutritional point of view (specific exams to be requested, timing of nutritional assessment etc).

In this context, the research team agrees with the recommendations previously published by Kikut et al. (Nutrients. 2020 Jun 30;12(7):1961) for the nutritional management of patients with neuroendocrine tumors, where it is proposed that standardizing nutritional care of patients with NETs should be a priority direction in the management of these patients. We have proposed specific exams to be requested and timing of nutritional assessment in page 14, line number 392.

- English language should be improved

Sorry for the inconveniences English may cause. We have rephrased some sentences and paragraphs aiming to improve the manuscript reading.

All the authors are very grateful for your corrections to substantially improve the quality of the manuscript.

Your sincerely.

Reviewer 2 Report

Dear Author,

This is an interesting paper. 

Here are my observations/questions/comments:

1.   Abstract/Summary – Please explain the abbreviations when first used (LDH, L3, ECOG)

2.    Introduction – the same as no.1 for “BMI” (before line 83), “DXA”

3.    Introduction – Lines 70-72 – A distinction between osteosarcopenic obesity and cancer – related sarcopenia should be done; they display different underling mechanisms, and sometimes overlap during lifespan of a certain individual

4.    Introduction – Lines 92-93 – This is true if we do not consider NET/NENs as cancer because, otherwise, there are numerous similar studies in traditional oncologic patients. I suggest you mention this distinction

5.    Methods – Line 96 - You got to the number (N=105) of enrolled patients after applying inclusion/exclusion criteria thus you should mention them first.  

6.    Results – Line 126 – “unknown”, not “un-known”

7.    Inclusion/exclusion criteria – Did you take into consideration associated endocrine diseases and iatrogenic outcomes that might lead to the misinterpretation of body composition (for instance, Cushing syndrome, acromegaly, glucocorticoid therapy, insulin therapy)?

8.    Figure 2B – typo “isulinoma”

9.    Table 2 – “diabetes” should be “diabetes mellitus”. Please comment what types of diabetes mellitus (type 1, type 2, secondary including iatrogenic) and which is the impact of it since one third of patients had this diagnostic that has an important impact on body composition. Did you have any cases of pre-diabetes (impaired fasting glucose or impaired glucose tolerance)?

10. By “carcinoid” do you mean serotonin producing tumours or NETs associated with carcinoid syndrome?

11. Table 2 – by “analogue” do you mean somatostatin analogues – octreotide and lanreotide?

12. Table 2 – “FA” is not explained as abbreviation

Best regards,

Author Response

Dear reviewer,

Thank you very much for all the suggestions you have made. In this letter, we respond to each of them. Changes made to the manuscript are marked in red.

REVIEWER 2:

  1. Abstract/Summary – Please explain the abbreviations when first used (LDH, L3, ECOG).

Thank you very much for all the comments. We have explained the abbreviations in the manuscript.

ECOG (Eastern Cooperative Oncology Group performance status).

LDH (lactate dehydrogenase).

L3 level (level of the third lumbar).

  1. Introduction – the same as no.1 for “BMI” (before line 83), “DXA”

Thank you very much for all the comments. We have explained the abbreviations in the manuscript.

BMI (body mass index)

Dual-energy X-ray absorptiometry (DXA)

  1. Introduction – Lines 70-72 – A distinction between osteosarcopenic obesity and cancer – related sarcopenia should be done; they display different underling mechanisms, and sometimes overlap during lifespan of a certain individual.

Thanks a lot for the suggestion. We have added and cited the difference between osteosarcopenic obesity and cancer-related sarcopenia in the text (page 2, line 75).

  1. Introduction – Lines 92-93 – This is true if we do not consider NET/NENs as cancer because, otherwise, there are numerous similar studies in traditional oncologic patients. I suggest you mention this distinction.

Thank you very much for the note. We have changed the structure of the introduction by adding your suggestions in the first paragraph (page 2, line 57):

“Although, on the contrary to most of other types of cancer, the majority of GEP-NET patients   usually have mild symptoms and progress over several years. However, some patients rapidly develop a severe disease. Despite different parameters that have been used to date3, there is still no feasible clinical or laboratory index that has been found as particularly useful to predict outcomes”

5.         Methods – Line 96 - You got to the number (N=105) of enrolled patients after applying inclusion/exclusion criteria thus you should mention them first.

We apologize for the misunderstanding. N changes due to inclusion/exclusion criterias were not clearly specified in text. We have written a better explanation now (page 3, line 104).

  1. Results – Line 126 – “unknown”, not “un-known”

Thanks for the notice. It has been changed (page 4, line 172).

  1. Inclusion/exclusion criteria – Did you take into consideration associated endocrine diseases and iatrogenic outcomes that might lead to the misinterpretation of body composition (for instance, Cushing syndrome, acromegaly, glucocorticoid therapy, insulin therapy)?

We completely agree. For this reason, we used these diseases as an exclusion criteria. We already specified it in the paper.

  1. Figure 2B – typo “isulinoma”

Thanks for the notice. It has been changed.

  1. Table 2 – “diabetes” should be “diabetes mellitus”. Please comment what types of diabetes mellitus (type 1, type 2, secondary including iatrogenic) and which is the impact of it since one third of patients had this diagnostic that has an important impact on body composition. Did you have any cases of pre-diabetes (impaired fasting glucose or impaired glucose tolerance)?

It is quite an interesting question. Type 2 diabetes mellitus (T2DM) correlates significantly to some body composition features as provided in the figure and the table below. However, it has no correlation with mortality (p=0.77) and body composition at the same time, so it was not considered as a confounding factor between the outcome and our interest variables (the body composition). This fact, then, was not further analyzed for the manuscript since it was not the aim of the study.

Concerning prediabetes, we cannot clarify the syndrome based on the available data. Besides, neither glucose nor T2DM have correlations to measured features, so we guess it would not have an effect on them. This results were added in results (page 10, line 269, supplementary table S2).

Measures (cm²/m²)

Overall

No T2DM

T2DM

p value

Test

Total body area

262.05 (64.24)

251.7895 (62.82453)

275.5544 (59.36912)

0.07663

T-test

Subcutaneous fat area

66.59 (33.12)

70.44305 (34.47183)

62.95904 (32.01727)

0.1497

Mann- Whitney wilcoxon

Visceral fat area

61.55 (38.72)

50.72197

(31.29403)

72.84834 (39.37159)

0.01505

Mann- Whitney Wilcoxon

Intermuscular fat area

133.76 (59.36)

5.230239 (4.155015)

5.928339 (3.094747)

0.1719

Mann- Whitney Wilcoxon

Total fat area

5.61 (3.82)

126.3953 (57.78653)

141.7357 (56.92119)

0.3159

Mann- Whitney Wilcoxon

Very low density muscle area

4.88 (2.73)

4.342222 (2.437471)

5.509409 (2.857705)

0.06624

Mann- Whitney Wilcoxon

Low density muscle area

13.27 (5.29)

12.16497 (4.95841)

14.59226 (4.924284)

0.06267

Mann- Whitney Wilcoxon

Normal density muscle area

23.67 (8.02)

24.82547 (8.381614)

21.66515 (6.909492)

0.05621

T-test

Total muscle area

42.33 (7.59)

41.87893 (7.905753)

42.22696 (7.21791)

0.45

Mann- Whitney Wilcoxon

Type 2 diabetes mellitus and body composition.

  1. By “carcinoid” do you mean serotonin producing tumours or NETs associated with carcinoid syndrome?

We apologize for the misunderstanding. We mean NETs associated with carcinoid syndrome. We have corrected the concept in the text (page 4, line 170). Thank you very much.

  1. Table 2 – by “analogue” do you mean somatostatin analogues – octreotide and lanreotide?

Yes, we mean somatostatin analogs such as lanreotide or octreotide (page 6, table 2). We apologize for the misunderstanding.  Thanks a lot for the comment.

  1. Table 2 – “FA” is not explained as abbreviation
    Thanks for the comment. It is atrial fibrillation. Already corrected.

All the authors are very grateful for your corrections to substantially improve the quality of the manuscript.

Your sincerely.
